# Advances in Fucoxanthin Research for the Prevention and Treatment of Inflammation-Related Diseases

**DOI:** 10.3390/nu14224768

**Published:** 2022-11-11

**Authors:** Biyun Guan, Kunsen Chen, Zhiyong Tong, Long Chen, Qi Chen, Jingqian Su

**Affiliations:** 1Fujian Key Laboratory of Innate Immune Biology, Biomedical Research Center of South China, College of Life Sciences, Fujian Normal University, Fuzhou 350117, China; 2Department of Neurosurgery & Neurocritical Care, Huashan Hospital, Fudan University, Shanghai 200040, China

**Keywords:** fucoxanthin, anti-inflammatory, antioxidant, anti-tumor, anti-obesity

## Abstract

Owing to its unique structure and properties, fucoxanthin (FX), a carotenoid, has attracted significant attention. There have been numerous studies that demonstrate FX’s anti-inflammatory, antioxidant, antitumor, and anti-obesity properties against inflammation-related diseases. There is no consensus, however, regarding the molecular mechanisms underlying this phenomenon. In this review, we summarize the potential health benefits of FX in inflammatory-related diseases, from the perspective of animal and cellular experiments, to provide insights for future research on FX. Previous work in our lab has demonstrated that FX remarkably decreased LPS-induced inflammation and improved survival in septic mice. Further investigation of the activity of FX against a wide range of diseases will require new approaches to uncover its molecular mechanism. This review will provide an outline of the current state of knowledge regarding FX application in the clinical setting and suggest future directions to implement FX as a therapeutic ingredient in pharmaceutical sciences in order to develop it into a treatment strategy against inflammation-associated disorders.

## 1. Introduction

Fucoxanthin (FX), a small compound with the molecular formula C_42_H_58_O_6_ and a molecular mass of 658.91 Da [1] (Figure 1), was first isolated from brown algae by Willstätter and Page in 1914, and its structure was later resolved by Englert et al. [2]. Unlike other carotenoids, including astaxanthin and beta-carotene, FX has a unique structure with several oxygen-containing functional groups (such as epoxy, hydroxyl, carbonyl, and carboxyl) and an unusual propadiene bond [3]. FX is the first alkene carotenoid identified in brown algae [4]. Owing to the propadiene bond and unstable chirality, FX is highly resistant to oxidation [5,6] yet susceptible to heat, air, and light degradation [7].

FX accounts for more than 10% of the estimated total carotenoid yield worldwide [8]. Natural carotenoids are commonly found in animals, plants, algae, and fungi but are only synthesized in plants, algae, and several species of fungi and bacteria [9,10]. As it is a carotenoid found in both brown diatoms and macroalgae, FX is significantly more abundant in aquatic environments than lutein or beta-carotene, both of which are commonly found in terrestrial crops [11,12]. As a key player in photosynthesis, FX absorbs light energy and protects chlorophyll from damage caused by excessive light [13].

FX is a safe pharmaceutical ingredient with minimal toxicity and side effects [14]. In rats, a rapid conversion of FX to FxOH was detected in their plasma 0.5 h after the oral administration of 65 mg/kg FX. The pharmacokinetic parameters of FxOH were as follows: the time at maximum concentration (T_max_) value was 11 h, the terminal half-life (t_1/2_) was 9.3 h, the maximum concentration (C_max_) was 263.3 μg/L, and the area under the curve (AUC_0-∞_) was 5304.2 μg·h/L [15]. In contrast, in humans, the pharmacokinetic parameters of FxOH were: T_max_ of 4 h, t_1/2_ of 7 h, C_max_ of 44.2 nmol/L, and AUC_0-∞_ of 663.7 nmol·h/L [16].

While FX can be chemically synthesized, it is relatively easier and more cost-effective to extract from certain microalgae and brown algae. As microalgae culture is expensive, almost all commercial FX has been derived from brown algae, despite microalgae having a higher FX content [8,17]. Recently, FX has been widely used in the cosmeceutical, food, and nutraceutical industries and has been shown to accelerate metabolism without stimulating the central nervous system or causing other significant side effects [18,19]. FX has several physiological properties, such as anti-inflammatory [20], antioxidant [21], anti-cancer [22], and anti-obesity [23] effects, which suggest a potential clinical application in the treatment of inflammation-related diseases [24].

## 2. Materials and Methods

To obtain an overview of the literature that covered FX, we searched the PubMed, Google Scholar, Elsevier, and CNKI databases for relevant reports. The keywords “fucoxanthin”, “carotenoid”, and “inflammation” were used separately and in combination with “diseases”, “anti-inflammatory”, “antioxidant”, “anti-tumor”, “cancer”, and “anti-obesity”. We focused on articles that were published in the last 15 years. Deduplication was performed after importing the resulting references into the Zotero reference management software.

## 3. Anti-Inflammatory Effects of FX

Inflammation is the body’s response against harmful stimuli to prevent tissue injury and restore homeostasis [25]. When the inflammatory response is accompanied by the increased production of white blood cells, there is a risk of developing a fever, swelling, and pain, as well as potentially disabling certain organs [20]. Simultaneously, the body produces a large number of cytokines, including interleukin (IL)-1, tumor necrosis factor (TNF), and IL-6, all of which are involved in regulating cell death in damaged tissues, altering vascular endothelial permeability, recruiting blood cells to affected tissues, and inducing acute phase protein production [26]. Inflammation is a dynamic process in which most cytokines are multifunctional, with some exerting both anti- and pro-inflammatory activities [27]. In the long term, cytokines mediate repair and remodeling by activating matrix metalloproteinases (MMPs) and collagen production, as well as by regulating integrins, angiogenesis, and progenitor cell mobilization. Inflammatory cytokines can have beneficial effects, such as healing, as well as detrimental effects, such as the induction of the chronic dilation of the heart, leading to heart failure [28].

Currently, studies on the mechanism of FX’s anti-inflammatory effects have mainly focused on NF-κB- and MAPK-associated pathways, along with other related signaling pathways (Figure 2). In RAW264.7 cells, FX reduced PGE2 and NO levels by inhibiting iNOS and COX-2 expression, as well as decreased IL-6, IL-1β, and TNF-α levels by inhibiting MAPK and NF-κB pathways [29,30]. Furthermore, Kim et al. found that, in RAW264.7 cells, FX inhibited lipopolysaccharide (LPS)-induced oxidative stress and inflammation by activating Nrf2 via the PI3K/AKT pathway [31]. Su et al. discovered that FX could inhibit IκB-α degradation in the cytoplasm, as well as inhibit the phosphorylation of NF-κB, thus suppressing the expression of inflammatory factors induced by LPS [32,33]. 

Moreover, in RAW264.7 cells, Li et al. reported that FX could attenuate the release of pro-inflammatory cytokines (IL-1β, IL-6, IL-10, TNF-α, iNOS, and COX-2) by inhibiting the TLR4/MyD88/NF-κB pathway, thereby preventing LPS-induced inflammation. The authors speculated that FX may inhibit TLR4 signaling by interacting with MD-2 [34]. The effects of FX on palmitate-induced inflammation have also been demonstrated to improve mitochondrial dysfunction and lipid metabolism while suppressing the expression of genes associated with M1 markers (Nlrp3, TNF-α, IL-1β, and IL-6) and upregulating those associated with the M2 marker (TGFβ1), thereby inhibiting macrophage-induced inflammation [35].

In an LPS-induced inflammation model, FX has been shown to inhibit endotoxin-induced uveitis in rats by inhibiting the protein expression of COX-2 and iNOS in vivo [36]. According to Li et al., in a mouse model of acute lung injury (ALI), FX significantly attenuated LPS-induced lung inflammation [34]. In addition, LPS is often used in acute inflammation studies in sepsis animal models [37]. Su et al. showed that FX significantly reduced LPS-induced inflammation and improved survival in septic mice [33], while another study found that FX alleviated the level of inflammatory expression in a mouse model of carrageenan-induced foot edema by inhibiting protein kinase B/Akt, NF-κB, and MAPK pathways [38]. In a mouse model of DSS-induced colitis, colitis was also improved following FX treatment, mainly by inhibiting the NF-κB/COX-2/PGE2 pathway [39]. In a mouse model of atopic dermatitis (AD), Chika et al. showed that FX normalized the immune response by modulating ILCreg while exerting anti-inflammatory effects on keratin-forming cells, thereby mitigating AD symptoms [40]. To evaluate the therapeutic effects of carotenoids on dinitrofluorobenzene-induced allergic reactions in mice, Sakai et al. noted that FX remarkably suppressed TNF-α and histamine levels as well as ear swelling. FX can also inhibit the biological effects of mast cell degranulation in the internal environment [41].

## 4. Antioxidant Effects of FX

The antioxidant activity of FX has been thoroughly investigated. During early free radical scavenging studies, FX was found to be effective in scavenging free radicals and capable of scavenging reactive oxygen species (ROS) [3,5].

In pathological processes, nitrogen species and ROS are mainly produced via aerobic metabolic processes and are involved in the pathological biochemical processes of degenerative diseases [42], as well as multiple regulatory effects on inflammation [43]. Not only do cytokines produce ROS, but they are also induced by ROS [44,45]. Carotenoids eliminate ROS through singlet molecular oxygen (^1^O_2_) and peroxyl radicals [42,46] and can clear free radicals in three steps: electron transfer, hydrogen extraction, and addition [46]. By accepting electrons from the reactants, conjugated double bonds enable these compounds to neutralize free radicals. [47].

FX has strong antioxidant activity because of its propylene structure with six oxygen atoms, making it highly sensitive to free radicals. The antioxidant capacity of FX is assessed by measuring its scavenging ability of free radicals, such as 1,1-diphenyl-2-picrylhydrazyl (DPPH), 12-doxyl-stearic acid (12-DS), the radical adduct of nitrosobenzene with linolenic acid radical (NB-L), 2,2′-azo-bis-2 amidinopropane (ABAP), 2,2′-Azino-bis(3-ethylbenzthiazoline-6-sulfonic acid) (ABTS), and 2,2′-azo-bis- (2-amidinopropane) dihydrochloride (AAPH) [5,48,49,50]. Nishino [51] reported that FX inhibits DPPH, 12-DS, and NB-L radicals by 28, 66, and 57%, respectively. The EC_50_ values of FX scavenging ABTS and DPPH are 0.03 and 0.14 mg/mL, respectively [52].

By enhancing the binding of nuclear factor erythroid 2-related factor (Nrf2) to the antioxidant response element in the promoter, FX protects cells from oxidative damage by enhancing the production of glutathione (GSH) (Figure 3). Nrf2 is activated when ROS are released, separating itself from cytoskeleton-associated proteins and translocating to the nucleus, where it activates antioxidant enzymes, preventing oxidative stress. [53].

As shown in Figure 3, FX could protect hepatic L02 cells against oxidative damage induced by H_2_O_2_, possibly via the PI3K-dependent activation of Nrf2 signaling, according to Wang et al. [54]. FX also prevents iron and arachidonic acid-induced oxidative damage in HepG2 cells, thereby reducing oxidation levels in the liver [55]. In SH-SY5Y cells, FX markedly reduces ROS levels and attenuates the neuronal loss induced by beta amyloid (Aβ) oligomers, possibly through the activation of the PI3K/Akt pathway and the inhibition of the ERK pathway [56]. In animal models, it has been demonstrated by Ha et al. that FX improved antioxidant capacity by activating the Nrf2 pathway and its downstream target gene, NQO1, in rats fed with a high-fat diet [57]. Yang et al. found that FX (25–50 mg/kg/day) reduced cadmium chloride (CdCl2)-induced apoptosis and oxidative stress in mice, resulting in normalization of the thyroid microstructure and ultrastructure. The authors concluded that FX had a protective effect against the cadmium-induced impairment of thyroid function, demonstrating the antioxidant properties of FX [58]. In addition, FX could improve cadmium-induced kidney injury by inhibiting apoptosis and oxidative stress and boosting mitochondrial structural integrity [59]. Zheng et al. found that FX minimized liver damage caused by alcohol in mice by activating the Nrf2-mediated antioxidant response [60].

Among all the body’s barriers, the skin is the first and largest, protecting our internal organs from the external environment. Excessive UV radiation leads to an increase in ROS, resulting in DNA damage, which can be further amplified by the expression of inflammatory cytokines, such as TNF-α, IL-8, IL-6, and IL-1β, in skin keratinocytes [61]; FX can absorb ultraviolet light (UVB 280–320 nm, UVA 320–400 nm) and visible light (VIS 400–700 nm) [62]. In vitro, Heo found that FX inhibits ROS formation in human fibroblasts following UVB radiation and increases cell survival, which demonstrates the protective effect of FX against UVB-induced cell damage [63]. Furthermore, FX can protect human HaCaT keratin-forming cells from UVB-induced damage [64]. 

Mio et al. also found that FX reduces ROS production in cells irradiated with UV, and the protective effect of FX on UV exposure-induced sunburn in mice was independent of direct UV absorption [65]. In vivo findings have also suggested that FX protects against UVB-induced skin photoaging and reduces UVB-induced epidermal hypertrophy and wrinkles [66]. Moreover, in hairless mice, FX cream reduces UVB-induced erythema by upregulating HO-1 and downregulating iNOS and COX-2 via the Nrf2 pathway, which could be used to prevent the deterioration associated with inflammatory skin pathology and protect against UV radiation [64].

## 5. Anti-Tumor and Anti-Cancer Effects of FX

In a report released by the World Health Organization (WHO) in 2020, cancer ranked as the number two cause of death worldwide, killing approximately 10 million individuals worldwide [67]. Despite available cancer treatment strategies, including surgical intervention, chemotherapy, radiation, and immunotherapy, cancer-related mortality remains high. Clinical and epidemiological findings have implicated inflammation as one of the primary causes of cancer [68]. Chronic inflammation is a key inducer of tumor development, which is referred to as “inflammation-associated carcinoma [69]”. Chronic, dysregulated, persistent, and unresolved inflammation is strongly linked to cancer development and malignant progression [70]. Rudolf Virchow first proposed the connection between inflammation and cancer in the 19th century, based upon the fact that cancer occurs in areas of chronic inflammation in the body and the detection of large numbers of inflammatory cells in tumor biopsies [71]. The association between cancer and inflammation has also been assessed by Dvorak, who found that tumors and wound tissue share similar stromal cell types—only tumors did not heal, while the wound eventually healed [72]. 

Research on FX anti-tumor and anti-cancer properties has advanced. Indeed, FX’s therapeutic effects have been demonstrated in animal or cellular models for highly prevalent cancers, such as liver, stomach, leukemia, breast, glioma, colon, cervical, and nasopharyngeal cancers (Figure 4) [1,22,73,74,75]. 

NF-κB plays a significant role in tumor development as a transcription factor in inflammation [76]. Abnormalities in inflammatory pathways, such as PI3K/AKT, JAK-STAT, MAPK, and NF-κB, cause the dysregulation of inflammatory factors [68]. Therefore, controlling the development of inflammation is an effective anti-cancer strategy. Due to the anti-inflammatory properties of FX, it may act as an anti-cancer agent. 

As shown in Figure 4, in vitro experimental results suggest that FX can inhibit inflammatory imbalance and reduce tumor-induced lymph angiogenesis in human breast cancer MDA-MB-231 cells [22] by decreasing the expression levels of p-PI3K, p-Akt, NF-κB, VEGF-C, and VEGF receptor-3 in human lymphatic endothelial cells, thereby inhibiting breast cancer cells through the regulation of inflammation. Ishikawa [19] reported that co-treatment with FX (10 μM) and its metabolite, Fucoxanthinol (FxOH) (5 μM), has a therapeutic effect against adult T-cell leukemia (ATL), a fatal T-lymphocyte malignancy caused by human T-cell leukemia virus type 1 (HTLV-1) infection. Mechanistically, FX and FxOH effects were mediated by the inactivation of AP-1 transcription factor (AP-1) or NF-ĸB, which induces GADD45α expression, cell cycle G1 phase arrest, and the reduction of cyclin D1/D2, CDK4/6, Bcl-2, XIAP, CIAP1/2, and surviving expression, thereby inducing apoptosis. In addition, FX treatment in HepG2 cancer cells can exacerbate toxicity and death through multiple pathways (apoptotic, antioxidant, and anti-inflammatory pathways) [77]. To regulate inflammation, FX can also control the development of malignancy by arresting the cell cycle and causing apoptosis. Treatment with FX (50 and 75 μM) has been shown to induce G2/M arrest and apoptosis in human gastric adenocarcinoma MGC-803 cells and significantly decrease the expression of CyclinB1, survivin, and STAT3 [78]. 

In vivo, using azomethanes/dextran sodium sulfate (AOM/DSS)-induced colorectal carcinoma mice, Terasaki et al. found that oral FX administration for 14 weeks (50 mg/kg body weight) prevented colorectal carcinogenesis [79]. Transcriptome analysis revealed that FX administration significantly decreased the activities of 11 signaling pathways, namely, adhesion, cell cycle, chemokine receptors, interleukins, Wnt, p53, MAPK, TGF-β, PI3K/AKT, STAT, and RAS. FX administration inhibited pancreatic tumorigenesis in mice models of pancreatic cancer, as reported by Murase et al. [80]. A significant alteration in the expression of 174 genes was identified in pancreatic tumors when FX was administered. The genetic analysis of protein expression, inflammation, and oncogenic effects has confirmed that the FX prevention of pancreatic cancer is directly linked to the inhibition of CCL21 and BTLA expression. Kim et al. observed a decrease in tumor size in mice implanted with B16F10 melanoma cells after receiving injections of FX (0.3 mg/mouse) intraperitoneally [81]. In addition, in nude mice implanted with HeLa cells, Ye et al. found that the gavage administration of FX (10 and 20 mg/kg) inhibited the growth of tumors through the PI3K/Akt pathway, along with a reduction in NF-κB activation [82]. 

## 6. Anti-Obesity Effects of FX

By the year 2030, obesity is projected to potentially affect more than 1 billion individuals [53]. Obesity significantly increases the risk of several diseases, including type II diabetes, stroke, sleep apnea, hypertension, high cholesterol, fatty liver disease, coronary atherosclerosis, and cancers, such as breast, endometrial, and colon cancer, thereby posing a significant burden on health care systems [83,84,85,86]. Therefore, it is essential to seek effective obesity preventive measures. As shown in Figure 5, current research advances suggest that FX can play a beneficial role against obesity through multiple pathways.

At the moment, obesity is thought to be caused by a homeostatic imbalance in adipose tissue brought on by an imbalance between the amount of energy consumed and the amount expended, which eventually results in chronic low-grade systemic inflammation (CLGSI) [87]. Obesity is also an associated cause of insulin resistance [88,89]. The regulation of cytokine secretion from abdominal adipocytes, the infiltration of macrophages into adipose tissue, and the induction of uncoupling protein 1 (UCP1) production in abdominal white adipose tissue (WAT) are important components in the prevention of obesity [88]. Adipocytes secrete several cytokines, such as leptin, resistin, lipocalin, or endolipin, while inflammatory cells infiltrating adipose tissue secrete monocyte chemotactic protein 1 (MCP-1), TNF-α, and IL-6 [90], all of which act on immune cells and induce systemic inflammation. In particular, MCP-1 releases into the bloodstream via abdominal WAT and elicits an inflammatory response in obese individuals by increasing the infiltration and activation of macrophages in adipose tissue [91], thereby increasing blood TNF-α and IL-6 levels while limiting glucose uptake in response to insulin, which facilitates insulin resistance. 

Chang et al. [92] investigated whether FX can inhibit lipid accumulation in FL83B hepatocytes. By establishing a model where FL83B cells are induced into fatty liver cells within 48 h using 0.5 mM oleic acid and then 24 h of treatment with various FX concentrations, by promoting the Sirt1/AMPK pathway, the authors demonstrated that FX increased lipolysis and inhibited adipogenesis in fatty liver cells. In 3T3-F442A adipocytes, it has also been reported that FxOH effectively inhibits TNF-α-induced mRNA overexpression and the protein secretion of MCP-1 and IL-6 [93]. In addition, FxOH decreases the mRNA expression of inducible nitric oxide synthase (iNOS), TNF-α, and cyclooxygenase 2 (COX-2) in palmitate-stimulated RAW264.7 macrophages [94]. FxOH effectively inhibited NO production, as well as IL-6, MCP-1, and PAI-1 mRNA expression, in a co-culture of adipocytes and macrophages stimulated with LPS [94]. Moreover, TLR4 receptors in macrophages are activated by free fatty acids produced in the adipocyte lipolysis of triglycerides in order to cause inflammation [95]. In this process, FxOH can regulate adipocyte inflammation via TLR4 receptors in macrophages. This suggests that FxOH can lessen the inflammation in adipocytes caused by obesity.

The effects of FX in vivo have been shown to attenuate obesity and related metabolic disorders induced by high-fat diets (HFDs) [96,97,98]. In addition, mildly elevated concentrations of IL-6 and TNF-α in patients with obesity have been reported [99]. FX treatment (100 mg/100 g diet) not only significantly reduced HFD-induced blood lipid levels, adipose tissue weight, liver weight, and weight gain but also inhibited mRNA and protein expression levels of the inflammatory factors IL-6 and TNF-α and increased the mRNA levels of the anti-inflammatory factor IL-10. This is significant because IL-10 overexpression can improve obesity and reduce excessive appetite [100,101]. FX reduces the expression of adipocyte cytokines and exerts anti-obesity effects by increasing the WAT expression of UCP1 [88]. UCP1 is normally found in brown adipose tissue, and in the absence of stimulation, WAT does not express it. Maeda et al. [102] showed that FX could modulate insulin signaling, improve glucose tolerance, and inhibit adipocyte-induced CLGSI in diabetic mice, while the WAT expression of pro-inflammatory genes, such as MCP-1 and TNF-α, was reduced. 

MCP-1 mRNA is highly expressed in mice fed with an HFD; feeding FX-rich wakame lipids normalized MCP-1 mRNA expression and upregulated UCP1 protein and mRNA expressions in WAT [97]. Furthermore, with increased UCP1 expression, dietary FX at 0.2% significantly attenuated WAT weight gain in KK-A^y^ mice [103]. 

Similarly, in humans, FX had a significant effect on weight loss. Indeed, the consumption of a mixture of 300 mg pomegranate seed oil and 300 mg brown seaweed extract containing 2.4 mg FX notably reduced obese women’s body weight and body and liver fat contents, and liver function tests were improved after 16 days of treatment [104]. Meanwhile, the daily intake of 3 mg FX displayed a positive weight loss effect in patients with obesity [105]. In this study, the authors administered capsules containing FX or placebo to Japanese adults with a body mass index (BMI) of >25 kg/m^2^ for 4 weeks. Before and after treatment, assessments of the weight, circumference of the neck, arms, and thighs, body composition, and abdominal fat area were performed. The FX-treated group had significantly lower relative values of body weight, BMI, total fat mass, subcutaneous fat area, waist circumference, and right thigh circumference than the placebo group. Moreover, blood pressure, blood parameters, pulse rate, and urinalysis parameters showed no abnormalities, and no significant side effects were observed. 

## 7. Conclusions

FX has received increasing attention as a special marine origin carotenoid with a high safety profile and various biological activities. In vitro and ex vivo research has shown that FX can reduce tissue damage and organ function impairment caused by excessive inflammatory responses and has great potential for the treatment and prevention of diseases that are related to inflammation, such as dermatitis, acute colitis, acute lung injury, sepsis, obesity, cancer, and diabetes. 

Research on FX in inflammatory disease treatment has shown promising results; however, the specific anti-inflammatory targets and signaling mechanisms of FX remain unclear and warrant further research. FX’s applications in the pharmaceutical, cosmetic, and food industries will be limited due to its low stability. Thus, a strategy to enhance the stability of FX will facilitate its use as a substitute for synthetic drugs in the pharmaceutical industry. Meanwhile, considering that FX metabolism differs in humans and animals, clinical trials are necessary to test the physiological advantages of FX in clinical settings to develop FX into an anti-inflammatory agent. 

## Figures and Tables

**Figure 1 nutrients-14-04768-f001:**
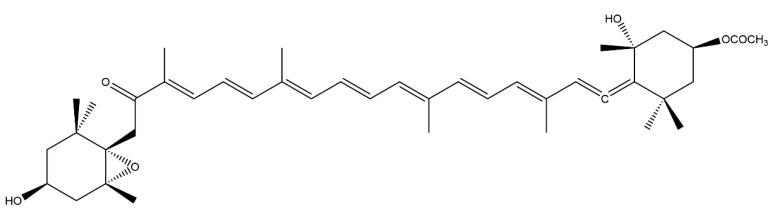
Chemical structure of fucoxanthin.

**Figure 2 nutrients-14-04768-f002:**
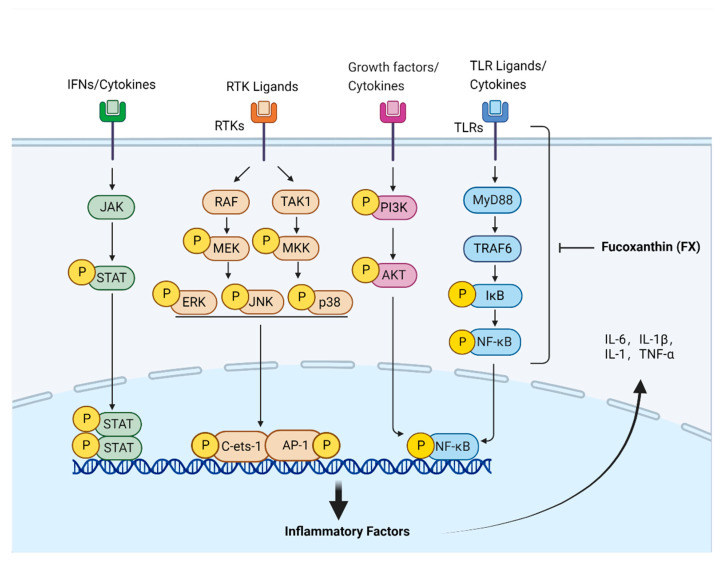
Pathway underlying fucoxanthin-associated anti-inflammatory effects. The development and progression of inflammation is accompanied by the activation of JAK/STAT, MAPK, PI3K/AKT, TLR4/MyD88/NF-κB and other related signaling pathways. FX can inhibit the signal transduction of these inflammatory pathways and down-regulate the expression of inflammatory factors such as IL-6, IL-1β, IL-1, and TNF-α, thereby achieving the anti-inflammatory effect.

**Figure 3 nutrients-14-04768-f003:**
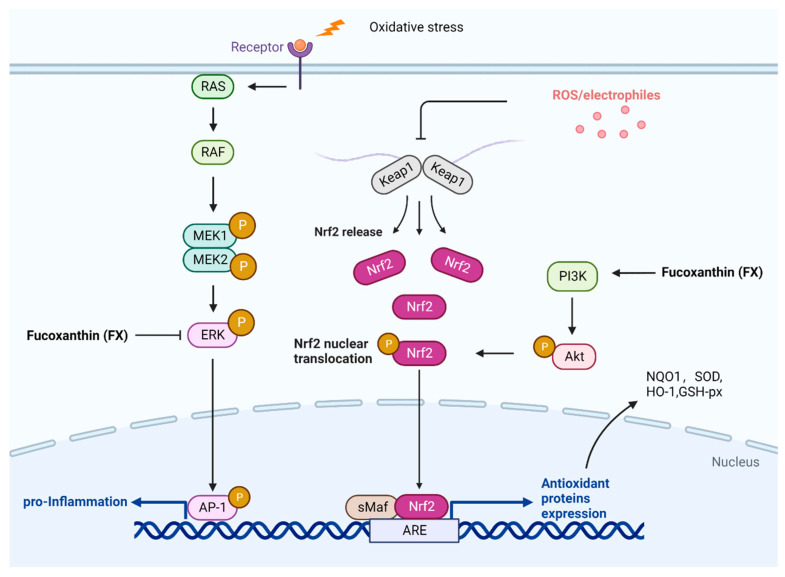
Pathway underlying the antioxidant effects of fucoxanthin (FX). Cellular oxidative stress leads to the expression of intracellular reactive oxygen species (ROS) and some inflammatory factors, resulting in cellular damage. FX can reduce the release of inflammatory factors by inhibiting the ERK pathway. Furthermore, FX activates Nrf2 via PI3K/Akt and upregulates antioxidant protein expression, such as NQO1, SOD, HO-1, and GSH-px, resulting in antioxidant activity.

**Figure 4 nutrients-14-04768-f004:**
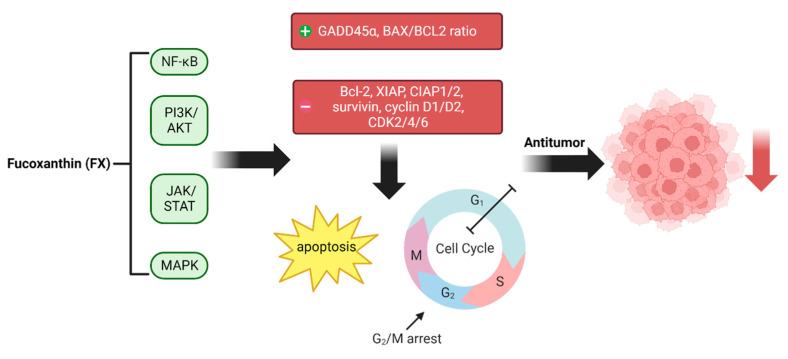
Pathway underlying the anti-tumor and anti-cancer effects of fucoxanthin (FX). FX achieves its anti-tumor effects by inhibiting inflammatory pathways such as NF-κB, MAPK, JAK-STAT, and PI3K/AKT. The expression level of GADD45α and the BAX/BCL2 ratio increased, and the expression of related cell cycle proteins, such as CDK2/4/6, Bcl-2, XIAP, and CIAP1/2, decreased after the related inflammatory pathways were inhibited by FX, which led to the regulation of apoptosis and cell cycle arrest, thus controlling tumor and cancer development and progression.

**Figure 5 nutrients-14-04768-f005:**
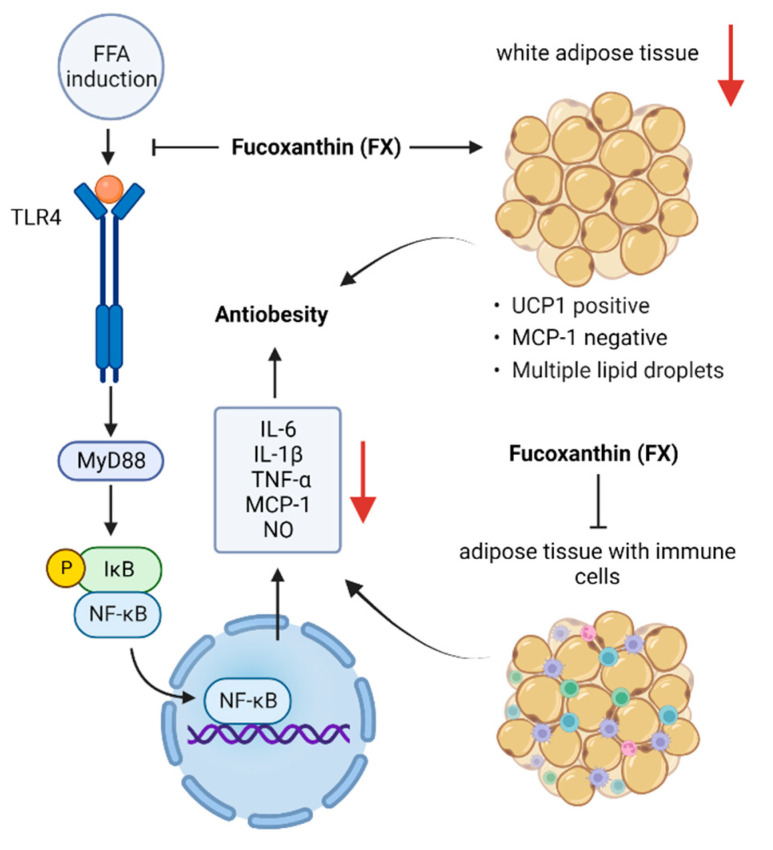
Pathway underlying the anti-obesity effects of fucoxanthin. Free fatty acids produced by adipocytes stimulate TLR4 receptors to induce inflammation. FX inhibits NF-κB entry into the nucleus and reduces the release of inflammatory factors, such as TNF-α, IL-6, and MCP-1, by modulating TLR4 receptors. FX also inhibits the release of inflammatory factors in adipocytes and macrophages infiltrating adipose tissue. Additionally, FX exerts anti-obesity effects by stimulating UCP-1 expression in white adipose tissue.

## Data Availability

Not applicable.

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
