# Peer review of "Advances in Fucoxanthin Research for the Prevention and Treatment of Inflammation-Related Diseases"

_nutrients, 2022, doi:10.3390/nu14224768_

Round 1

Reviewer 1 Report

Overall comments:

The authors present a review entitled: “Advances in fucoxanthin research for the prevention and treatment of inflammation-related diseases” in which many biological mechanisms and effects of fucoxanthin for many illnesses are reviewed. The article is written in good English. The methodology of the review could not be evaluated since there is no methodology section. The authors managed to write a short but informative review, which I believe aggregates relevant information about the effects and mechanisms of fucoxanthin. There already are many review articles of fucoxanthyn in the recent 5 years, however, I believe the manuscript brings good discussion in regards to the molecular signaling pathways. The only major review concern for the article is that there is no review methodology section after the introduction and the conclusion needs to highlight more the uniqueness of this review in comparison to the many others already available. Review articles, even literature reviews, must display their review methodology in a specific section (e.g. Name of the database searched, terms used in the search, date range, number of withdrawn articles, filtering of articles and so forth) (the authors can look for the Prisma guidelines for reviews, in order to have more information, alas, I believe that the information that I have written in the previous “e.g.” is more than sufficient to understand your review methodology).

Specific comments:

L18-19

“development and application of FX” reevaluate and elaborate this sentence

L306-328 Conclusion

 What about the molecular effects dynamics reviewed by the paper? Were not those some of the main objectives of the review?

Author Response

Maria Luz Fernandez

Lluis Serra-Majem

Editor-in-Chief

Nutrients

Re: Manuscript ID: nutrients-1995770

Dear Editor,

Attached is the revised version of our manuscript “Advances in fucoxanthin research for the prevention and treatment of inflammation-related diseases”, which we would like to resubmit for consideration to be published as a review article in your journal.

Your comments and those of the reviewers were highly insightful and enabled us to improve the quality of our manuscript. In the following pages are our point-by-point responses to each of the comments of the reviewer. Revisions in the text are highlighted using the “Track Changes”. We hope that the revisions in the manuscript and our accompanying responses would be sufficient to make our manuscript suitable for publication in Nutrients.

We shall look forward to hearing from you at your earliest convenience.

Yours sincerely,

Jingqian Su, Ph.D.

Associate Professor

Fujian Key Laboratory of Innate Immune Biology,

Biomedical Research Center of South China,

College of Life Science,

Fujian Normal University, Fuzhou 350117, Fujian, China

Tel: +86-18950498937

Responses to the comments of Reviewer #1

1.The only major review concern for the article is that there is no review methodology section after the introduction and the conclusion needs to highlight more the uniqueness of this review in comparison to the many others already available.

Response:

Many thanks for the reviewer suggestion. We are grateful for the suggestion. In the revised manuscript, we added the review methodology section as follow:

  1. Materials and Methods

     To get an overview of the literature that covered FX, we searched PubMed, Google Scholar, Elsevier, and CNKI databases for relevant reports. The keywords “fucoxanthin”, “carotenoid”, and “inflammation” were used separately and in combination with “dis-eases”, “anti-inflammatory”, “antioxidant”, “anti-tumor”, “cancer”, and “anti-obesity”. We focused on articles that were published in the last 15 years. Deduplication was per-formed after importing the resulting references into Zotero reference management soft-ware.

2.on Line 18-19:development and application of FX” reevaluate and elaborate this sentence

Response:

Many thanks for the reviewer suggestion.

The to provide insights into the development and application of FX” has been corrected to “to provide insights for future research on FX” in the line 18.

3. Line 306-328 Conclusion: What about the molecular effects dynamics reviewed by the paper? Were not those some of the main objectives of the review?

Response:

In fact, that is a very good point. As the reviewer indicated, the content of the revisions in the text are highlighted using the “Track Changes”. In particular, the content of the molecular effects dynamics has been revised in the line 48-54as follow:

In rats, a rapid conversion of FX to FxOH was detected in their plasma 0.5 h after oral administration of 65 mg/kg FX. The pharmacokinetic parameters of FxOH were as follows: the time at maximum concentration (Tmax) value was 11 h, terminal half-life (t1/2) was 9.3 h, maximum concentration (Cmax) was 263.3 μg/L, and area under the curve (AUC0-∞) was 5304.2 μg·h/L [15]. In contrast, in humans, the pharmacokinetic parameters of FxOH were: Tmax of 4 h, t1/2 of 7 h, Cmax of 44.2 nmol/L, and AUC0-∞ of 663.7 nmol·h /L [16].

Reviewer 2 Report

Guan B and colleagues have provided a very detailed overview of the current knowledge on the potential benefits of fucoxanthin (FX) for human health. This review summarizes the beneficial activity of fucoxanthin as an anti-inflammatory, anti-oxidant agent with comprehensive analysis literature and citation of the studies. The authors also discuss the collection of in vitro, in vivo, and clinical studies to provide a current scenario of the understanding of the therapeutic potential of fucoxanthin as an anti-tumor and anti-obesity agent.

My only comment for the authors is to provide the detailed figure legends for Fig 2-5. In my opinion, this will allow readers to easily understand the possible target or pathways involved in the proposed action of fucoxanthin.  

Author Response

Maria Luz Fernandez

Lluis Serra-Majem

Editor-in-Chief

Nutrients

Re: Manuscript ID: nutrients-1995770

Dear Editor,

Attached is the revised version of our manuscript “Advances in fucoxanthin research for the prevention and treatment of inflammation-related diseases”, which we would like to resubmit for consideration to be published as a review article in your journal.

Your comments and those of the reviewers were highly insightful and enabled us to improve the quality of our manuscript. In the following pages are our point-by-point responses to each of the comments of the reviewer. Revisions in the text are highlighted using the “Track Changes”. We hope that the revisions in the manuscript and our accompanying responses would be sufficient to make our manuscript suitable for publication in Nutrients.

We shall look forward to hearing from you at your earliest convenience.

Yours sincerely,

Jingqian Su, Ph.D.

Associate Professor

Fujian Key Laboratory of Innate Immune Biology,

Biomedical Research Center of South China,

College of Life Science,

Fujian Normal University, Fuzhou 350117, Fujian, China

Tel: +86-18950498937

Responses to the comments of Reviewer #2

1. My only comment for the authors is to provide the detailed figure legends for Fig 2-5. In my opinion, this will allow readers to easily understand the possible target or pathways involved in the proposed action of fucoxanthin.

Response:

Many thanks for the reviewer suggestion. As the reviewer indicated, the detailed figure legends have been added to Figure 2-5 as follow:

Figure 2. Pathway underlying fucoxanthin-associated anti-inflammatory effects. The development and progression of inflammation is accompanied by the activation of JAK/STAT, MAPK, PI3K/AKT, TLR4/MyD88/NF-κB and other related signaling pathways. FX can inhibit the signal transduction of these inflammatory pathways and down-regulate the ex-pression of inflammatory factors such as IL-6, IL-1β, IL-1, and TNF-α, thereby achieving the anti-inflammatory effect.

Figure 3. Pathway underlying the antioxidant effects of fucoxanthin (FX). Cellular oxidative stress leads to the expression of intracellular reactive oxygen species (ROS) and some inflammatory factors, resulting in cellular damage. FX can reduce the re-lease of inflammatory factors by inhibiting the ERK pathway. Furthermore, FX activates Nrf2 via PI3K/Akt and upregulates antioxidant protein expression, such as NQO1, SOD, HO-1 and GSH-px, resulting in antioxidant activity.

Figure 4. Pathway underlying the anti-tumor and anti-cancer effects of fucoxanthin (FX). FX achieves its anti-tumor effects by inhibiting inflammatory pathways such as NF-κB, MAPK, JAK-STAT, and PI3K/AKT. The expression level of GADD45α and the BAX/BCL2 ratio increased and the expression of related cell cycle proteins, such as CDK2/4/6, Bcl-2, XIAP, and CIAP1/2, decreased after the related inflammatory pathways were inhibited by FX, which led to the regulation of apoptosis and cell cycle arrest; thus, controlling tumor and cancer development and progression.

Figure 5. Pathway underlying the anti-obesity effects of fucoxanthin. Free fatty acids produced by adipocytes stimulate TLR4 receptors to induce inflammation. FX inhibits NF-κB entry into the nucleus and reduces the release of inflammatory factors, such as TNF-α, IL-6, and MCP-1, by modulating TLR4 receptors. FX also inhibits the release of inflam-matory factors in adipocytes and macrophages infiltrating adipose tissue. Additionally, FX ex-erts anti-obesity effects by stimulating UCP-1 expression in white adipose tissue.

Round 2

Reviewer 1 Report

The suggestions were satisfactorily attended to. I recommend the paper for publication.